# Secure Multi-party Differential Privacy

**Peter Kairouz**[1]  **Sewoong Oh**[2]  **Pramod Viswanath**[1]

[1]Department of Electrical & Computer Engineering
[2]Department of Industrial & Enterprise Systems Engineering
University of Illinois Urbana-Champaign
Urbana, IL 61801, USA
{kairouz2,swoh,pramodv}@illinois.edu

## Abstract

We study the problem of interactive function computation by multiple parties, each possessing a bit, in a differential privacy setting (i.e., there remains an uncertainty in any party's bit even when given the transcript of interactions and all the other parties' bits). Each party wants to compute a function, which could differ from party to party, and there could be a central observer interested in computing a separate function. Performance at each party is measured via the accuracy of the function to be computed. We allow for an arbitrary cost metric to measure the distortion between the true and the computed function values. Our main result is the optimality of a simple non-interactive protocol: each party randomizes its bit (sufficiently) and shares the privatized version with the other parties. This optimality result is very general: it holds for all types of functions, heterogeneous privacy conditions on the parties, all types of cost metrics, and both average and worst-case (over the inputs) measures of accuracy.

## 1  Introduction

Multi-party computation (MPC) is a generic framework where multiple parties share their information in an *interactive* fashion towards the goal of computing some functions, potentially different at each of the parties. In many situations of common interest, the key challenge is in computing the functions as *privately* as possible, i.e., without revealing much about one's information to the other (potentially colluding) parties. For instance, an interactive voting system aims to compute the majority of (say, binary) opinions of each of the parties, with each party being averse to declaring their opinion publicly. Another example involves banks sharing financial risk exposures – the banks need to agree on quantities such as the overnight lending rate which depends on each bank's exposure, which is a quantity the banks are naturally loath to truthfully disclose [1]. A central learning theory question involves characterizing the fundamental limits of interactive information exchange such that a strong (and suitably defined) adversary only learns as little as possible while still ensuring that the desired functions can be computed as accurately as possible.

One way to formulate the privacy requirement is to ensure that each party learns nothing more about the others' information than can be learned from the output of the function computed. This topic is studied under the rubric of *secure function evaluation* (SFE); the SFE formulation has been extensively studied with the goal of characterizing which functions can be securely evaluated [39, 3, 21, 11]. One drawback of SFE is that depending on what auxiliary information the adversary might have, disclosing the exact function output might reveal each party's data. For example, consider computing the average of the data owned by all the parties. Even if we use SFE, a party's data can be recovered if all the other parties collaborate. To ensure protection of the private data under such a strong adversary, we want to impose a stronger privacy guarantee of differential privacy. Recent breaches of sensitive information about individuals due to linkage attacks prove

the vulnerability of existing ad-hoc privatization schemes, such as anonymization of the records. In linkage attacks, an adversary matches up anonymized records containing sensitive information with public records in a different dataset. Such attacks have revealed the medical record of a former governor of Massachusetts [37], the purchase history of Amazon users[7], genomic information [25], and movie viewing history of Netflix users [33].

An alternative formulation is differential privacy, a relatively recent formulation that has received considerable attention as a formal mathematical notion of privacy that provides protection against such strong adversaries (a recent survey is available at [16]). The basic idea is to introduce enough randomness in the communication so that an adversary possessing arbitrary side information and access to the entire transcript of the communication will still have some residual uncertainty in identifying any of the bits of the parties. This privacy requirement is strong enough that non-trivial functions will be computed only with some error. Thus, there is a great need for understanding the fundamental tradeoff between privacy and accuracy, and for designing privatization mechanisms and communication protocols that achieve the optimal tradeoffs. The formulation and study of an optimal framework addressing this tradeoff is the focus of this paper.

We study the following problem of multi-party computation under differential privacy: each party possesses a single bit of information and the information bits are statistically independent. Each party is interested in computing a function, which could differ from party to party, and there could be a central observer (observing the entire transcript of the interactive communication protocol) interested in computing a separate function. Performance at each party and the central observer is measured via the accuracy of the function to be computed. We allow an arbitrary cost metric to measure the distortion between the true and the computed function values. Each party imposes a differential privacy constraint on its information bit (the privacy level could be different from party to party) – i.e., there remains an uncertainty in any specific party's bit even to an adversary that has access to the transcript of interactions and all the other parties' bits. The interactive communication is achieved via a broadcast channel that all parties and the central observer can hear (this modeling is without loss of generality – since the differential privacy constraint protects against an adversary that can listen to the entire transcript, the communication between any two parties might as well be revealed to all the others). It is useful to distinguish between two types of communication protocols: *interactive* and *non-interactive*. We say a communication protocol is non-interactive if a message broadcasted by one party does not depend on the messages broadcasted by other parties. In contrast, interactive protocols allow the messages at any stage of the communication to depend on all the previous messages.

Our main result is the exact optimality of a simple non-interactive protocol in terms of maximizing accuracy for any given privacy level, when each party possesses one bit: each party randomizes (sufficiently) and publishes its own bit. In other words:

> *non-interactive randomized response is exactly optimal.*

Each party and the central observer then separately compute their respective decision functions to maximize the appropriate notion of their accuracy measure. This optimality result is very general: it holds for all types of functions, heterogeneous privacy conditions on the parties, all types of cost metrics, and both average and worst-case (over the inputs) measures of accuracy. Finally, the optimality result is *simultaneous*, in terms of maximizing accuracy at each of the parties and the central observer. Each party only needs to know its own desired level of privacy, its own function to be computed, and its measure of accuracy. Optimal data release and optimal decision making is naturally separated.

The key technical result is a geometric understanding of the space of conditional probabilities of a given transcript: the interactive nature of the communication constrains the space to be a rank-1 tensor (a special case of Equation (6) in [35] and perhaps implicitly used in [30]; the two-party analog of this result is in [29]), while differential privacy imposes linear constraints on the singular vectors of this tensor. We characterize the convex hull of such manifolds of rank-1 tensors and show that their corner-points exactly correspond to the transcripts that arise from a non-interactive randomized response protocol. This universal (for all functionalities) characterization is then used to argue that both average-case and worst-case accuracies are maximized by non-interactive randomized responses.

Technically, we prove that non-interactive randomized response is the optimal solution of the rank-constrained and non-linear optimization of (11). The rank constraints on higher order tensors arises from the necessary condition of (possibly interactive) multi-party protocols, known as protocol compatibility (see Section 2 for details). To solve this non-standard optimization, we transform (11) into a novel linear program of (17) and (20). The price we pay is the increased dimension, the resulting LP is now infinite dimensional. The idea is that we introduce a new variable for each possible rank-one tensor, and optimize over all of them.

Formulating utility maximization under differential privacy as linear programs has been previously studied in [32, 20, 6, 23], under the standard client-server model where there is a single data publisher and a single data analyst. These approaches exploit the fact that both the differential privacy constraints and the utilities are linear in the matrix representing a privatization mechanism. A similar technique of transforming a non-linear optimization problem into an infinite dimensional LP has been successfully applied in [26], where optimal privatization mechanisms under local differential privacy has been studied. We generalize these techniques to rank-constrained optimizations.

Further, perhaps surprisingly, we prove that this infinite dimensional linear program has a simple optimal solution, which we call *randomized response*. Upon receiving the randomized responses, each party can compute the best approximation of its respective function. The main technical innovation is in proving that $(a)$ the optimal solution of this LP corresponds to corner points of a convex hull of a particular manifold defined by a rank-one tensor (see Lemma 6.2 in the supplementary material for details); and $(b)$ the respective manifold has a simple structure such that the corner points correspond to particular protocols that we call randomized responses.

When the accuracy is measured via average accuracy, both the objective and the constraints are linear and it is natural to expect the optimal solution to be at the corner points (see Equation (17)). A surprising aspect of our main result is that the optimal solution is still at the corner points even though the worst-case accuracy is a concave function over the protocol $P$ (see Equation (19)).

This work focuses on the scenario where each party possesses a single bit of information. With multiple bits of information at each of the parties, the existence of a differentially private protocol with a fixed accuracy for any non-trivial functionality implies the existence of a protocol with the same level of privacy and same level of accuracy for a specific functionality that only depends on one bit of each of the parties (as in [22]). Thus, if we can obtain lower bounds on accuracy for functionalities involving only a single bit at each of the parties, we obtain lower bounds on accuracy for all non-trivial general functionalities. However, non-interactive communication is unlikely to be exactly optimal in this general case where each party possesses multiple bits of information, and we provide a further discussion in Section 4. We move a detailed discussion of related work (Section 5) to the supplementary material, focusing on the problem formulation next.

## 2   Problem formulation

Consider the setting where we have $k$ parties, each with its own private binary data $x_i \in \{0, 1\}$ generated independently. The independence assumption here is necessary because without it each party can learn something about others, which violates differential privacy, even without revealing any information. We discuss possible extensions to correlated sources in Section 4. Differential privacy implicitly imposes independence in a multi-party setting. The goal of the private multi-party computation is for each party $i \in [k]$ to compute an arbitrary function $f_i : \{0, 1\}^k \to \mathcal{Y}$ of interest by interactively broadcasting messages, while preserving the privacy of each party. There might be a central observer who listens to all the messages being broadcasted, and wants to compute another arbitrary function $f_0 : \{0, 1\} \to \mathcal{Y}$. The $k$ parties are honest in the sense that once they agree on what protocol to follow, every party follows the rules. At the same time, they can be curious, and each party needs to ensure other parties cannot learn his bit with sufficient confidence. The privacy constraints here are similar to the local differential privacy setting studied in [13] in the sense that there are multiple privacy barriers, each one separating each individual party and the rest of the world. However, the main difference is that we consider multi-party computation, where there are multiple functions to be computed, and each node might possess a different function to be computed.

Let $x = [x_1, \ldots, x_k] \in \{0, 1\}^k$ denote the vector of $k$ bits, and $x_{-i} = [x_1, \ldots, x_{i-1}, x_{i+1}, \ldots, x_k] \in \{0, 1\}^{k-1}$ is the vector of bits except for the $i$-th bit. The parties

agree on an interactive protocol to achieve the goal of multi-party computation. A 'transcript' is the output of the protocol, and is a random instance of all broadcasted messages until all the communication terminates. The probability that a transcript $\tau$ is broadcasted (via a series of interactive communications) when the data is $x$ is denoted by $P_{x,\tau} = \mathbb{P}(\tau \,|\, x)$ for $x \in \{0,1\}^k$ and for $\tau \in \mathcal{T}$. Then, a protocol can be represented as a matrix denoting the probability distribution over a set of transcripts $\mathcal{T}$ conditioned on $x$: $P = [P_{x,\tau}] \in [0,1]^{2^k \times |\mathcal{T}|}$.

In the end, each party makes a decision on what the value of function $f_i$ is, based on its own bit $x_i$ and the transcript $\tau$ that was broadcasted. A decision rule is a mapping from a transcript $\tau \in \mathcal{T}$ and private bit $x_i \in \{0,1\}$ to a decision $y \in \mathcal{Y}$ represented by a function $\hat{f}_i(\tau, x_i)$. We allow randomized decision rules, in which case $\hat{f}_i(\tau, x_i)$ can be a random variable. For the central observer, a decision rule is a function of just the transcript, denoted by a function $\hat{f}_0(\tau)$.

We consider two notions of accuracy: the average accuracy and the worst-case accuracy. For the $i$-th party, consider an accuracy measure $w_i : \mathcal{Y} \times \mathcal{Y} \to \mathbb{R}$ (or equivalently a negative cost function) such that $w_i(f_i(x), \hat{f}_i(\tau, x_i))$ measures the accuracy when the function to be computed is $f_i(x)$ and the approximation is $\hat{f}_i(\tau, x_i)$. Then the average accuracy for this $i$-th party is defined as

$$\text{ACC}_{\text{ave}}(P, w_i, f_i, \hat{f}_i) \quad \equiv \quad \frac{1}{2^k} \sum_{x \in \{0,1\}^k} \mathbb{E}_{\hat{f}_i, P_{x,\tau}}[w_i(f_i(x), \hat{f}_i(\tau, x_i))] \,, \tag{1}$$

where the expectation is taken over the random transcript $\tau$ distribution as $P$ and also any randomness in the decision function $\hat{f}_i$. We want to emphasize that the input is deterministic, i.e. we impose no distribution on the input data, and the expectation is *not* over the data sets $x$. Compared to assuming a distribution over the data, this is a weaker assumption on the data, and hence makes our main result stronger. For example, if the accuracy measure is an indicator such that $w_i(y, y') = \mathbb{I}_{(y=y')}$, then $\text{ACC}_{\text{ave}}$ measures the average probability of getting the correct function output. For a given protocol $P$, it takes $(2^k |\mathcal{T}|)$ operations to compute the optimal decision rule:

$$f^*_{i,\text{ave}}(\tau, x_i) \quad = \quad \arg \max_{y \in \mathcal{Y}} \sum_{x_{-i} \in \{0,1\}^{k-1}} P_{x,\tau} \, w_i(f_i(x), y) \,, \tag{2}$$

for each $i \in [k]$. The computational cost of $(2^k |\mathcal{T}|)$ for computing the optimal decision rule is unavoidable in general, since that is the inherent complexity of the problem: describing the distribution of the transcript requires the same cost. We will show that the optimal protocol requires a set of transcripts of size $|\mathcal{T}| = 2^k$, and the computational complexity of the decision rule for general a function is $2^{2^k}$. However, for a fixed protocol, this decision rule needs to be computed only once before any message is transmitted. Further, it is also possible to find a closed form solution for the decision rule when $f$ has a simple structure. One example is the XOR function studied in detail in Section 3, where the optimal decision rule is as simple as evaluating the XOR of all the received bits, which requires $O(k)$ operations. When there are multiple maximizers $y$, we can choose arbitrarily, and it follows that there is no gain in randomizing the decision rule for average accuracy. Similarly, the worst-case accuracy is defined as

$$\text{ACC}_{\text{wc}}(P, w_i, f_i, \hat{f}_i) \quad \equiv \quad \min_{x \in \{0,1\}^k} \mathbb{E}_{\hat{f}_i, P_{x,\tau}}[w_i(f_i(x), \hat{f}_i(\tau, x_i))] \,. \tag{3}$$

For worst-case accuracy, given a protocol $P$, the optimal decision rule of the $i$-th party with a bit $x_i$ can be computed by solving the following convex program:

$$Q^{(x_i)} = \arg \max_{Q \in \mathbb{R}^{|\mathcal{T}| \times |\mathcal{Y}|}} \min_{x_{-i} \in \{0,1\}^{k-1}} \sum_{\tau \in \mathcal{T}} \sum_{y \in \mathcal{Y}} P_{x,\tau} \, w_i(f_i(x), y) Q_{\tau, y} \tag{4}$$

$$\text{subject to} \quad \sum_{y \in \mathcal{Y}} Q_{\tau, y} = 1 \,, \ \forall \tau \in \mathcal{T} \text{ and } Q \geq 0$$

The optimal (random) decision rule $f^*_{i,\text{wc}}(\tau, x_i)$ is to output $y$ given transcript $\tau$ according to $\mathbb{P}(y|\tau, x_i) = Q^{(x_i)}_{\tau, y}$. This can be formulated as a linear program with $(|\mathcal{T}| |\mathcal{Y}|)$ variables and $(2^k + |\mathcal{T}|)$ constraints. Again, it is possible to find a closed form solution for the decision rule when $f$ has a simple structure: for the XOR function, the optimal decision rule is again evaluating

the XOR of all the received bits requiring $O(k)$ operations. For a central observer, the accuracy measures are defined similarly, and the optimal decision rule is now

$$f_{0,\text{ave}}^*(\tau) \quad = \quad \arg\max_{y \in \mathcal{Y}} \sum_{x \in \{0,1\}^k} P_{x,\tau}\, w_0(f_0(x), y)\,, \tag{5}$$

and for worst-case accuracy the optimal (random) decision rule $f_{0,\text{wc}}^*(\tau)$ is to output $y$ given transcript $\tau$ according to $\mathbb{P}(y|\tau) = Q_{\tau,y}^{(0)}$. Subject to $\sum_{y \in \mathcal{Y}} Q_{\tau,y} = 1$, $\forall \tau \in \mathcal{T}$ and $Q \geq 0$,

$$Q^{(0)} = \arg\max_{Q \in \mathbb{R}^{|\mathcal{T}| \times |\mathcal{Y}|}} \quad \min_{x \in \{0,1\}^k} \sum_{\tau \in \mathcal{T}} \sum_{y \in \mathcal{Y}} P_{x,\tau}\, w_0(f_0(x), y) Q_{\tau,y} \tag{6}$$

where $w_0 : \mathcal{Y} \times \mathcal{Y} \to \mathbb{R}$ is the measure of accuracy for the central observer.

Privacy is measured by differential privacy [14, 15]. Since we allow heterogeneous privacy constraints, we use $\varepsilon_i$ to denote the desired privacy level of the $i$-th party. We say a protocol $P$ is $\varepsilon_i$-differentially private for the $i$-th party if for $i \in [k]$, and all $x_i, x_i' \in \{0,1\}$, $x_{-i} \in \{0,1\}^{k-1}$, and $\tau \in \mathcal{T}$,

$$\mathbb{P}(\tau|x_i, x_{-i}) \quad \leq \quad e^{\varepsilon_i}\, \mathbb{P}(\tau|x_i', x_{-i})\,. \tag{7}$$

This condition ensures no adversary can infer the private data $x_i$ with high enough confidence, no matter what auxiliary information he might have and independent of his computational power. To lighten notations, we let $\lambda_i = e^{\varepsilon_i}$ and say a protocol is $\lambda_i$-differentially private for the $i$-th party. If the protocol is $\lambda_i$-differentially private for all $i \in [k]$, then we say that the protocol is $\{\lambda_i\}$-differentially private for all parties.

A necessary condition on the multi-party protocols $P$, when the bits are generated independent of each other, is protocol compatibility [22]: conditioned on the transcript of the protocol, the input bits stay independent of each other. In our setting, input bits are deterministic, hence independent. Mathematically, a protocol $P$ is protocol compatible if each column $P^{(\tau)}$ is a *rank-one tensor*, when reshaped into a $k$-th order tensor $P^{(\tau)} \in [0,1]^{2 \times 2 \times \cdots \times 2}$, where

$$P_{x_1,\ldots,x_k}^{(\tau)} \quad = \quad P_{x,\tau}\,. \tag{8}$$

Precisely, there exist vectors $u^{(1)} \ldots, u^{(k)}$ such that $P^{(\tau)} = u^{(1)} \otimes \cdots \otimes u^{(k)}$, where $\otimes$ denotes the standard outer-product, i.e. $P_{i_1,\ldots,i_k}^{(\tau)} = u_{i_1}^{(1)} \times \cdots \times u_{i_k}^{(k)}$. This is crucial in deriving the main results, and it is a well-known fact in the secure multi-party computation literature. This follows from the fact that when the bits are generated independently, all the bits are still independent conditioned on the transcript, i.e. $P(x|\tau) = \prod_i P(x_i|\tau)$, which follows implicitly from [30] and directly from Equation (6) of [35]. Notice that using the rank-one tensor representation of each column of the protocol $P^{(\tau)}$, we have $P(\tau|x_i = 0, x_{-i})/P(\tau|x_i = 1, x_{-i}) = u_1^{(i)}/u_2^{(i)}$. It follows that $P$ is $\lambda_i$-differentially private if and only if $\lambda_i^{-1} u_2^{(i)} \leq u_1^{(i)} \leq \lambda_i u_2^{(i)}$.

**Randomized response.** Consider the following simple protocol known as the *randomized response*, which is a term first coined by [38] and commonly used in many private communications including the multi-party setting [31]. We will show in Section 3 that this is the optimal protocol for simultaneously maximizing the accuracy of all the parties. Each party broadcasts a randomized version of its bit denoted by $\tilde{x}_i$ such that

$$\tilde{x}_i = \begin{cases} x_i & \text{with probability } \frac{\lambda_i}{1+\lambda_i}\,, \\ \bar{x}_i & \text{with probability } \frac{1}{1+\lambda_i}\,, \end{cases} \tag{9}$$

where $\bar{x}_i$ is the logical complement of $x_i$. Each transcript can be represented by the output of the protocol, which in this case is $\tilde{x} = [\tilde{x}_1, \ldots, \tilde{x}_k] \in \mathcal{T}$, where $\mathcal{T} = \{0,1\}^k$ is now the set of all broadcasted bits.

**Accuracy maximization.** Consider the problem of maximizing the average accuracy for a centralized observer with function $f$. Up to the scaling of $1/2^k$ in (1), the accuracy can be written as

$$\sum_{x \in \{0,1\}^k} \mathbb{E}_P[w(f(x), \hat{f}_0(\tau))] \quad = \quad \sum_x \sum_{y \in \mathcal{Y}} \underbrace{w(f_0(x), y)}_{\triangleq W_x^{(y)}} \sum_{\tau \in \mathcal{T}} P_{x,\tau} \underbrace{\mathbb{P}(\hat{f}_0(\tau) = y)}_{\triangleq Q_{\tau,y}}\,, \tag{10}$$

where $\hat{f}_0(\tau)$ denotes the randomized decision up on receiving the transcript $\tau$. In the following we define $W_x^{(y)} \triangleq w(f_0(x), y)$ to represent the accuracy measure and $Q_{\tau,y} \triangleq \mathbb{P}(\hat{f}(\tau) = y)$ to represent the decision rule.

Focusing on this single central observer for the purpose of illustration, we want to design protocols $P_{x,\tau}$ and decision rules $Q_{\tau,y}$ that maximize the above accuracy. Further, this protocol has to be compatible with interactive communication, satisfying the rank one condition discussed above, and satisfy the differential privacy condition in (7). Hence, we can formulate the accuracy maximization can be formulated as follows. Given $W_x^{(y)}$'s in terms of the function $f_0(\cdot)$ to be computed, an accuracy measure $w_0(\cdot, \cdot)$, and required privacy level $\lambda_i$'s, we solve

$$\underset{P \in \mathbb{R}^{2^k \times |\mathcal{T}|}, Q \in \mathbb{R}^{|\mathcal{T}| \times |\mathcal{Y}|}}{\text{maximize}} \quad \sum_{x, \in \{0,1\}^k, y \in \mathcal{Y}} W_x^{(y)} \sum_{\tau \in \mathcal{T}} P_{x,\tau} Q_{\tau,y},$$

subject to $\quad P$ and $Q$ are row-stochastic matrices, $\quad \text{rank}(P^{(\tau)}) = 1, \; \forall \tau \in \mathcal{T},$

$$P_{(x_i, x_{-i}), \tau} \le \lambda_i P_{(x_i', x_{-i}), \tau}, \quad \forall i \in [k], x_1, x_1', \in \{0,1\}, x_{-i} \in \{0,1\}^{k-1}, \tag{11}$$

where $P^{(\tau)}$ is defined as a $k$-th order tensor defined from the $\tau$-th column of matrix $P$ as defined in Equation (8). Notice that the rank constraint is only a necessary condition for a protocol to be compatible with interactive communication schemes, i.e. a valid interactive communication protocol implies the rank-one condition but not all rank-one protocols are valid interactive communication schemes. Therefore, the above is a relaxation with larger feasible set of protocols, but it turns out that the optimal solution of the above optimization problem is the randomized response, which is a valid (non-interactive) communication protocol. Hence, there is no loss in solving the above relaxation.

The main challenge in solving this optimization is that it is a rank-constrained tensor optimization which is notoriously difficult. Since the rank constraint is over a $k$-th order tensor ($k$-dimensional array) with possibly $k > 2$, common approaches of convex relaxation from [36] for matrices (which are 2nd order tensors) does not apply. Further, we want to simultaneously apply similar optimizations to all the parties with different functions to be computed.

We introduce a novel transformation of the above rank-constrained optimization into a linear program in (17) and (20). The price we pay is in the increased dimensionality: the LP has an infinite dimensional decision variable. However, combined with the geometric understanding of the the the manifold of rank-1 tensors, we can identify the exact optimal solution. We show in the next section that given desired level of privacy $\{\lambda_i\}_{i \in [k]}$, there is a single universal protocol that simultaneously maximizes the accuracy for $(a)$ all parties; $(b)$ any functions of interest; $(c)$ any accuracy measures; and $(d)$ both worst-case and average case accuracy. Together with optimal decision rules performed at each of the receiving ends, this gives the exact optimal multi-party computation scheme.

## 3  Main Result

We show, perhaps surprisingly, that the simple randomized response presented in (9) is the optimal protocol in a very general sense. For any desired privacy level $\lambda_i$, and arbitrary function $f_i$, for any accuracy measure $w_i$, and any notion of accuracy (either average or worst case), we show that the randomized response is universally optimal. The proof of the following theorem can be found in the supplementary material.

**Theorem 3.1** *Let the optimal decision rule be defined as in* (2) *for the average accuracy and* (4) *for the worst-case accuracy. Then, for any $\lambda_i \ge 1$, any function $f_i : \{0,1\}^k \to \mathcal{Y}$, and any accuracy measure $w_i : \mathcal{Y} \times \mathcal{Y} \to \mathbb{R}$ for $i \in [k]$, the randomized response for given $\lambda_i$ with the optimal decision function achieves the maximum accuracy for the $i$-th party among all $\{\lambda_i\}$-differentially private interactive protocols and all decision rules. For the central observer, the randomized response with the optimal decision rule defined in* (5) *and* (6) *achieves the maximum accuracy among all $\{\lambda_i\}$-differentially private interactive protocols and all decision rules for any arbitrary function $f_0$ and any measure of accuracy $w_0$.*

This is a strong universal optimality result. Every party and the central observer can *simultaneously* achieve the optimal accuracy using a universal randomized response. Each party only needs to know

its own desired level of privacy, its own function to be computed, and its measure of accuracy. Optimal data release and optimal decision making are naturally separated. However, it is not immediate at all that a non-interactive scheme such as the randomized response would achieve the maximum accuracy. The fact that interaction does not help is counter-intuitive, and might as well be true only for the binary data scenario we consider in this paper. The key technical innovation is the convex geometry in the proof, which does not generalize to larger alphabet case.

Once we know interaction does not help, we can make an educated guess that the randomized response should dominate over other non-interactive schemes. This intuition follows from the dominance of randomized response in the single-party setting, that was proved using a powerful operational interpretation of differential privacy first introduced in [34]. This intuition can in fact be made rigorous, as we show in Section 7 (of our supplemental material) with a simple two-party example. However, we want to emphasize that our main result for multi-party computation does not follow from any existing analysis of randomized responses, in particular those seemingly similar analyses in [26]. The challenge is in proving that interaction does not help, which requires the technological innovations presented in this paper.

**Multi-party XOR computation.** For a given function and a given accuracy measure, analyzing the performance of the optimal protocol provides the exact nature of the privacy-accuracy tradeoff. Consider a scenario where a central observer wants to compute the XOR of all the $k$-bits, each of which is $\lambda$-differentially private. In this special case, we can apply our main theorem to analyze the accuracy exactly in a combinatorial form, and we provide a proof in Section A.1.

**Corollary 3.1** *Consider $k$-party computation for $f_0(x) = x_1 \oplus \cdots \oplus x_k$, and the accuracy measure is one if correct and zero if not, i.e. $w_0(0,0) = w_0(1,1) = 1$ and $w_0(0,1) = w_0(1,0) = 0$. For any $\{\lambda\}$-differentially private protocol $P$ and any decision rule $\hat{f}$, the average and worst-case accuracies are bounded by*

$$\mathrm{ACC}_{\mathrm{ave}}(P, w_0, f_0, \hat{f}_0) \;\leq\; \frac{\sum_{i=0}^{\lfloor k/2 \rfloor} \binom{k}{2i} \lambda^{k-2i}}{(1+\lambda)^k} \;,\quad \mathrm{ACC}_{\mathrm{wc}}(P, w_0, f_0 \hat{f}_0) \;\leq\; \frac{\sum_{i=0}^{\lfloor k/2 \rfloor} \binom{k}{2i} \lambda^{k-2i}}{(1+\lambda)^k} \;,$$

*and the equality is achieved by the randomized response and optimal decision rules in* (5) *and* (6).

The optimal decision for both accuracies is simply to output the XOR of the received privatized bits. This is a strict generalization of a similar result in [22], where XOR computation was studied but only for a two-party setting. In the high privacy regime, where $\varepsilon \simeq 0$ (equivalently $\lambda = e^\varepsilon \simeq 1$), this implies that $\mathrm{ACC}_{\mathrm{ave}} = 0.5 + 2^{-(k+1)} \varepsilon^k + O(\varepsilon^{k+1})$ . The leading term is due to the fact that we are considering an accuracy measure of a Boolean function. The second term of $2^{-(k+1)} \varepsilon^k$ captures the effect that, we are essentially observing the XOR through $k$ consecutive binary symmetric channels with flipping probability $\lambda/(1+\lambda)$. Hence, the accuracy gets exponentially worse in $k$. On the other hand, if those $k$-parties are allowed to collaborate, then they can compute the XOR in advance and only transmit the privatized version of the XOR, achieving accuracy of $\lambda/(1+\lambda) = 0.5 + (1/4)\varepsilon^2 + O(\varepsilon^3)$. This is always better than not collaborating, which is the bound in Corollary 3.1.

## 4  Discussion

In this section, we discuss a few topics, each of which is interesting but non-trivial to solve in any obvious way. Our main result is general and sharp, but we want to ask how to push it further.

**Generalization to multiple bits.** When each party owns multiple bits, it is possible that interactive protocols improve over the randomized response protocol. This is discussed with examples in Section 8 (in the supplementary material).

**Approximate differential privacy.** A common generalization of differential privacy, known as the approximate differential privacy, is to allow a small slack of $\delta \geq 0$ in the privacy condition[14, 15]. In the multi-party context, a protocol $P$ is $(\varepsilon_i, \delta_i)$-differentially private for the $i$-th party if for all $i \in [k]$, and all $x_i, x_i' \in \{0,1\}$, $x_{-i} \in \{0,1\}^{k-1}$, and for all subset $T \subseteq \mathcal{T}$,

$$\mathbb{P}(\tau \in T | x_i, x_{-i}) \;\leq\; e^{\varepsilon_i} \mathbb{P}(\tau \in T | x_i', x_{-i}) + \delta_i \;. \tag{12}$$

It is natural to ask if the linear programming (LP) approach presented in this paper can be extended to identify the optimal multi-party protocol under $\{(\varepsilon_i, \delta_i)\}$-differential privacy. The LP formulations of (17) and (20) heavily rely on the fact that any differentially private protocol $P$ can be decomposed as the combination of the matrix $S$ and the $\theta^{(y)}$'s. Since the differential privacy constraints are invariant under scaling of $P_\tau^{(y)}$, one can represent the scale-free pattern of the distribution with $S_\tau$ and the scaling with $\theta_\tau^{(y)}$. This is no longer true for $\{(\varepsilon_i, \delta_i)\}$-differential privacy, and the analysis technique does not generalize.

**Correlated sources.** When the data $x_i$'s are correlated (e.g. each party observe a noisy version of the state of the world), knowing $x_i$ reveals some information on other parties' bits. In general, revealing correlated data requires careful coordination between multiple parties. The analysis techniques developed in this paper do not generalize to correlated data, since the crucial rank-one tensor structure of $S_\tau^{(y)}$ is no longer present.

**Extensions to general utility functions.** A surprising aspect of the main result is that even though the worst-case accuracy is a concave function over the protocol $P$, the maximum is achieved at an extremal point of the manifold of rank-1 tensors. This suggests that there is a deeper geometric structure of the problem, leading to possible universal optimality of the randomized response for a broader class of utility functions. It is an interesting task to understand the geometric structure of the problem, and to ask what class of utility functions lead to optimality of the randomized response.

## Acknowledgement

This research is supported in part by NSF CISE award CCF-1422278, NSF SaTC award CNS-1527754, NSF CMMI award MES-1450848 and NSF ENG award ECCS-1232257.

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
