[Supplementary Material]

# 5 Related Work

In the context of two parties, privacy-accuracy tradeoffs have been studied in [31, 22] where a single function is computed by a "third-party" observing the transcript of the interactive protocol. [31] constructs natural functions that can only be computed very coarsely (using a natural notion of accuracy) as compared to a client-server model (which is essentially the single party setting). [22] shows that every non-trivial boolean functionality always incurs some loss of accuracy for any non-trivial privacy setting. Further, focusing on the specific scenario where each of the two parties has a single bit of information, [22] characterizes the *exact* accuracy-privacy tradeoff for AND and XOR functionalities; the corresponding optimal protocol turns out to be *non-interactive*. However, this result was derived under some assumptions: only two parties are involved, only the central observer computes an approximation of a function, the function has to be either XOR or AND, symmetric privacy conditions were used for both of the parties, and accuracy was measured only as worst-case over the four possible inputs. Further, their analysis technique does not generalize to the case when we have more than two parties. To this end, we provide a new analysis technique of transforming the rank constrained optimization problem into a linear program, and give the exact optimal protocols for any number of parties, any function of interest, heterogeneous privacy requirements, and both average and worst-case accuracy measures. Among other things, this fully recovers the main results of [22] and does it with a more efficient protocol as discussed in Section 7.

While there is a vast literature on differential privacy in a variety of contexts, exact optimality results are very few. In an early result, [20] shows that adding discrete Laplacian noise to scalar count queries (which are a special case of integer functionalities with sensitivity one) is universally optimal in terms of maximizing the average accuracy for any cost metric that is monotonic in the error. While such universal mechanisms do not exist in terms of maximizing average accuracy [6], recent work by [18] and [19] construct a class of mechanisms (termed as "staircase" mechanisms) that are universally optimal in terms of maximizing worst-case accuracy for any cost metric that is monotonic in the error. Demonstrating a fundamental equivalence between binary hypothesis testing and differential privacy, [34] derives data processing inequalities for differential privacy that are used to derive optimal composition theorems (characterization of how privacy degrades due to interactive querying). These techniques are also useful in the results derived in this paper.

The study of accuracy-privacy tradeoffs in the MPC context was first initiated by [2] (addressed in a more general context earlier by [17]) who studied a specific paradigm where differential privacy and SFE co-exist: the function to compute is decided from differentially private analyses and the method to compute it is decided from SFE theory. Specific functions such as SUM were studied in this setting, but no exact optimality results are available. Exact optimality of non-interactive communication is demonstrated for two-party AND and XOR function computations in [22]. A curious fact in the context of AND computation is that [22] requires the randomization of the bit to be in an output space of *three* letters (as opposed to the binary alphabet in standard randomized response). At a first glance, this appears to be in contradiction to the claim in this paper. A closer look reveals that randomized response also achieves the same performance (worst-case accuracy over the four inputs) when combined with a different (and randomized) decision function. Indeed, the techniques from [34] allow one to foresee this from an abstract point of view: every differentially private mechanism of a bit can be *simulated* from the output of randomized response with the same level of privacy. In other words, if $b$ is the bit, and $X$ is the (random) output of randomized response and $Y$ is the (random) output of some differentially private mechanism operating on $b$, then there exists a joint distribution on $(X, Y)$ such that the Markov chain $b - X - Y$ holds. This is discussed in detail in a later discussion section.

Function approximation has been widely studied in differential privacy literature under a centralized model where there is a single trusted entity owning a statistical database over a large number of individuals. Under this centralized setting, statistical learning has also been widely studied in differential privacy, e.g. classification [28, 10], k-means clustering [5], principal component analysis [8, 9, 24, 27]. In particular, it has been shown in [28] that under the centralized setting there exists a class of concepts that is efficiently learnable by *interactive* algorithms whereas a non-interactive algorithm requires exponential number of samples. An algorithm is called interactive in the centralized model, if it involves multiple rounds of communications between the server and the client. In contrast, we consider a multi-party setting where privacy barrier is on each individual owning his/her own data. All communication happens in multiple rounds in multi-party computation, and a protocol is called interactive in the multi-party setting if one party's message depends on other party's previous messages. In this sense, the notion of interaction in multi-party computation is significantly different from what has been previously studied under centralized client-server settings.

# 6 Proof of Theorem 3.1

We first focus on the scenario where a central observer wants to compute a function $f$ over $k$ bits distributed across $k$ parties. We will show in Section 6.1 that $\text{ACC}_{\text{ave}}(P, w, f, \hat{f})$ is maximized when randomized response protocol is used with the optimal decision rule of (5). Subsequently in Section 6.2, we show that

$\mathrm{ACC}_{\mathrm{wc}}(P, w, f, \hat{f})$ is maximized when again randomized response protocol is used with the optimal decision rule of (6). Theorem 3.1 directly follows from these two results, since the $i$-th party can compute the optimal decision and achieve the maximum accuracy for each instance of $x_i \in \{0, 1\}$.

## 6.1 Proof for the average case

**Theorem 6.1** *For a central observer who wants to compute $f$ with accuracy measure $w$, randomized response with the optimal decision rule of* (5) *maximizes the average accuracy* $\mathrm{ACC}_{\mathrm{ave}}(P, w, f, \hat{f})$ *among all* $\{\lambda_i\}$-*differentially private protocols and all decision rules.*

In this section, we provide a proof of this theorem. We want to solve the rank-constrained optimization problem of (11). The sketch of the proof is as follows. First, we introduce a novel change of variables to transform the optimization into an infinite dimensional linear program. Next, we show that if the optimal solution to this LP has non-zero probability only for 'extremal' transcripts (see Definition 6.1), then there is only one possible protocol which is the randomized response in (9). Finally, we finish the proof by using dual LP to prove that the optimal solution can only have non-zero probability at the 'extremal' transcripts.

**LP formulation.** We want to maximize the average accuracy over $P$ and $Q$, where the average accuracy is (up to a scaling by $1/2^k$)

$$\sum_x \mathbb{E}_P[w(f(x), \hat{f}(\tau))] = \sum_x \sum_{y \in \mathcal{Y}} \underbrace{w(f(x), y)}_{\triangleq W_x^{(y)}} \sum_{\tau \in \mathcal{T}} P_{x,\tau} \underbrace{\mathbb{P}(\hat{f}(\tau) = y)}_{\triangleq Q_{\tau, y}} = \sum_y \left\langle W^{(y)}, \sum_\tau P_\tau Q_{\tau, y} \right\rangle,$$

where $\langle , \rangle$ denote the standard inner product such that $\left\langle W^{(y)}, P_\tau Q_{\tau, y} \right\rangle = \sum_x \left( W_x^{(y)} P_{x,\tau} Q_{\tau, y} \right)$, and $P_\tau$ is the column of the matrix $P$ corresponding to $\tau$. The $2^k \times |\mathcal{T}|$-dimensional matrix $P$ represents the conditional distribution of the transcripts $\tau$ given the original data $x$, such that $P_{x,\tau} = \mathbb{P}(\tau|x)$. The $|\mathcal{T}| \times |\mathcal{Y}|$-dimensional matrix $Q$ represents the decision rule, possibly randomized. For example, if we consider two-party XOR computation with the same level of privacy $\lambda$, a solution (which turns out to be optimal) is randomized response with decision rule according to the XOR of the received bits. In particular, $\tau \in \{00, 01, 10, 11\}$ and $y = \hat{f}(\tau)$ is the XOR of the two bits in $\tau$. This can be written as

$$P = \frac{1}{(1 + \lambda)^2} \begin{bmatrix} \lambda^2 & \lambda & \lambda & 1 \\ \lambda & \lambda^2 & 1 & \lambda \\ \lambda & 1 & \lambda^2 & \lambda \\ 1 & \lambda & \lambda & \lambda^2 \end{bmatrix} , \quad \text{and } Q = \begin{bmatrix} 1 & 0 \\ 0 & 1 \\ 0 & 1 \\ 1 & 0 \end{bmatrix} . \tag{13}$$

Notice that the labeling of $\tau$ is arbitrary and applying the same permutation to the columns of $P$ and the rows of $Q$ does not change the feasibility or the accuracy of the solution. The columns of $P$ are still rank one when written in an appropriate tensor form, and also satisfy the differential privacy constraints. Another important point is that we cannot restrict the number of transcripts a priori, and when solving (11), we need to consider infinite dimensional (but countable) $\mathcal{T} = \mathbb{Z}$. The objective and the constraints depend on

$$[\mathbb{P}(y, \hat{f}(\tau) = y|x)]_{x, \tau, y} = [P_{x,\tau} Q_{\tau, y}]_{x, \tau, y} ,$$

for $x \in \{0, 1\}^k$, $\tau \in \mathbb{Z}$, and $y \in \mathcal{Y}$ where how we label or index the transcript $\tau$ is arbitrary. Since the rank constraints on the tensorized version of the columns of $P$ are difficult to handle, we exploit the fact that the problem is invariant in renaming of the transcript index $\tau$, and introduce a new indexing of the transcripts and new representation of the effective decision variable $[\mathbb{P}(y, \hat{f}(\tau) = y|x)]_{x, \tau, y}$.

Define a *signature vector* as a vector $S_{(s_1, \ldots, s_k)} \in \mathbb{R}^{2^k}$ indexed by $(s_1, \ldots, s_k) \in [\lambda_1^{-1}, \lambda_1] \times \cdots \times [\lambda_k^{-1}, \lambda_k]$. A signature vector $S_{s_1, \ldots, s_k}$ is a vectorized version of a rank-one tensor $[1, s_1] \otimes \cdots \otimes [1, s_k]$ (to ensure that the rank constraint is satisfied) with $\lambda_i^{-1} \leq s_i \leq \lambda_i$ for all $i \in [k]$ (to ensure that the differential privacy constraint is satisfied). The index $(s_1, \ldots, s_k)$ effectively replaces the indexing of the transcript $\tau$. Consider an infinite dimensional matrix $S$, where the number of rows is $2^k$ and the number of columns is uncountably infinite. The *signature matrix $S$* contains as its columns all possible choices of the signature vector $S_{(s_1, \ldots, s_k)}$ indexed by $(s_1, \ldots, s_k)$. Given this definition $S$, the space of all possible feasible protocols and all possible corresponding decision rules can be represented as

$$[\mathbb{P}(y, \hat{f}(\tau) = y|x)]_{x, \tau, y} = [S_{x, (s_1, \ldots, s_k)} \theta_{(s_1, \ldots, s_k)}^{(y)}]_{x, (s_1, \ldots, s_k), y} , \tag{14}$$

where the equality is up to a appropriate mapping of indexes in $\tau$ and $(s_1, \ldots, s_k)$ and merging/splitting/dropping of appropriate columns. As a concrete example, the conditional distribution of outputting

$y = 0$ in (13) is

$$[\mathbb{P}(\tau, \hat{f}(\tau) = y|x)]_{x,\tau,y=0} \;=\; P \operatorname{diag}(Q^{(0)}) \;=\; \frac{1}{(1+\lambda)^2} \begin{bmatrix} \lambda^2 & 0 & 0 & 1 \\ \lambda & 0 & 0 & \lambda \\ \lambda & 0 & 0 & \lambda \\ 1 & 0 & 0 & \lambda^2 \end{bmatrix}, \qquad (15)$$

which can be represented (up to a reindexing of the columns) using the signature matrix as

$$S \operatorname{diag}(\theta^{(0)}) \;=\; \begin{bmatrix} 1 & \cdots & 1 & \cdots \\ \lambda^{-1} & \cdots & \lambda & \cdots \\ \lambda^{-1} & \cdots & \lambda & \cdots \\ \lambda^{-2} & \cdots & \lambda^2 & \cdots \end{bmatrix} \begin{bmatrix} \frac{\lambda^2}{(1+\lambda)^2} & & & & & \\ & 0 & & & & \\ & & \ddots & & & \\ & & & 0 & & \\ & & & & \frac{1}{(1+\lambda)^2} & \\ & & & & & 0 \\ & & & & & & \ddots \end{bmatrix} \qquad (16)$$

For all practical purposes, these two matrices represent the same protocol and the same decision rule. Since $S$ is a fixed matrix for given problem parameters $k$ and $\lambda_i$'s, the new decision variable is just the set of scaling vectors $\{\theta^{(y)}\}_{y\in\mathcal{Y}}$. By optimizing over $\theta^{(y)}$'s, we are effectively selecting a subset of signatures to include in our transcript, and choosing the randomized outputs of those selected transcripts. We want to maximize the average accuracy, conditioned on the fact that conditional probabilities sum to one and probabilities are non-negative.

$$\begin{aligned} \underset{\theta^{(1)},\ldots,\theta^{(|\mathcal{Y}|)}}{\text{maximize}} \quad & \sum_{y\in\mathcal{Y}} \langle W^{(y)}, S\theta^{(y)} \rangle \;=\; \sum_{y\in\mathcal{Y},(s_1,\ldots,s_k)} (S^T W^{(y)})_{(s_1,\ldots,s_k)} \theta^{(y)}_{(s_1,\ldots,s_k)} \\ \text{subject to} \quad & \sum_{y\in\mathcal{Y}} \sum_{(s_1,\ldots,s_k)} S_{(s_1,\ldots,s_k)} \theta^{(y)}_{(s_1,\ldots,s_k)} = \mathbb{1} \\ & \theta^{(y)} \geq 0. \end{aligned} \qquad (17)$$

This is a linear program in $\theta^{(y)}$'s and once we have the optimal solution we can translate it to the original variables using (14). However, numerically solving the above problem is infeasible since the dimension of each variable $\theta^{(y)}$ is now uncountably infinite. We first claim that the solution of this problem is simple and can be represented in a closed form, and then prove this claim using the dual LP.

**Definition** A $2^k$-dimensional column vector $S_{(s_1,\ldots,s_k)}$ is *extremal* if the $k$-th order tensorization of $S_{(s_1,\ldots,s_k)}$ is a rank-one tensor of the form $[1 \ , \ s_1] \otimes \cdots \otimes [1 \ , \ s_k]$ with factors $s_i \in \{\lambda_i^{-1}, \lambda_i\}$ for all $i \in [k]$. There are $2^k$ such extremal columns of $S$.

This notion of extremal transcript is consistent with a similar notion of extremal privatization mechanisms defined in [26] as a set of mechanisms whose conditional distributions are at the extreme points of differential privacy constraints. When $k = 2$ there are four extremal columns of $S$:

$$\begin{bmatrix} 1 \\ \lambda_1 \\ \lambda_2 \\ \lambda_1\lambda_2 \end{bmatrix}, \quad \begin{bmatrix} 1 \\ \lambda_1 \\ \lambda_2^{-1} \\ \lambda_1\lambda_2^{-1} \end{bmatrix}, \quad \begin{bmatrix} 1 \\ \lambda_1^{-1} \\ \lambda_2 \\ \lambda_1^{-1}\lambda_2 \end{bmatrix}, \quad \begin{bmatrix} 1 \\ \lambda_1^{-1} \\ \lambda_2^{-1} \\ \lambda_1^{-1}\lambda_2^{-1} \end{bmatrix}.$$

We make the following claim.

**Remark** The optimal solution to the LP in (17) only has strictly positive $\theta^{(y)}_{(s_1,\ldots,s_k)}$ for $(s_1,\ldots,s_k)$ corresponding to extremal columns of $S$ and all the non-extremal columns are set to zero.

Suppose for now that this claim is true, then we can make following observations.

- There is an optimal solution of the LP that requires no randomized decision. Suppose the set $\{\theta^{(y)}\}_{y\in\mathcal{Y}}$ is an optimal solution, and there is an extremal transcript $(s_1,\ldots,s_k)$ such that both $\theta^{(y_1)}_{(s_1,\ldots,s_k)}$ and $\theta^{(y_2)}_{(s_1,\ldots,s_k)}$ are non-zero for some $y_1, y_2 \in \mathcal{Y}$. Then, we can construct a new optimal solution by setting $\tilde{\theta}^{(y_1)}_{(s_1,\ldots,s_k)} = \theta^{(y_1)}_{(s_1,\ldots,s_k)} + \theta^{(y_2)}_{(s_1,\ldots,s_k)}$ and $\tilde{\theta}^{(y_2)}_{(s_1,\ldots,s_k)} = 0$. Continuing in this fashion, we can construct an optimal solution with no randomization.

- Since the $2^k \times 2^k$ sub matrix of $S$ corresponding to the extremal columns is now an invertible matrix, $\theta = \sum_{y \in \mathcal{Y}} \theta^{(y)}$ is easily computed by the equality constraint. Once the optimal $\theta$ is fixed, we can identify the optimal decision rule for each transcript separately. Among $\theta^{(y)}_{(s_1,\ldots,s_k)}$'s for $y \in \mathcal{Y}$, put all the mass on the $y$ that maximizes $(S^T W^{(y)})_{(s_1,\ldots,s_k)}$. The optimal protocol $S \operatorname{diag}(\theta)$ is uniquely determined, and finding the optimal decision rule (i.e. $\theta^{(y)}$) is also simple once we have the protocol. This gives the precise optimal decision rule described in Equation (2).

- This uniquely determined optimal protocol is the randomized response defined in Equation (9) for all possible choices of the problem parameters, and it is a non-interactive protocol.

**Proof of the remark 6.1 using the geometry of the manifold of rank one tensors.** Now, we are left to prove the claim that the optimal solution only contains the extremal signatures. Consider a $k$-dimensional manifold in $2^k$-dimensional space:

$$
\begin{aligned}
\mathcal{M}_{\{\lambda_i\}} &= \{T : T = [1, t_1] \otimes \cdots \otimes [1, t_k] \text{ and } \lambda_i^{-1} \le t_i \le \lambda_i \text{ for all } i \in [k]\}, \\
\mathcal{P}_{\{\lambda_i\}} &= \operatorname{conv}(\mathcal{M}_{\{\lambda_i\}}),
\end{aligned}
$$

where $\operatorname{conv}(\cdot)$ is the convex hull of a set. The following result characterizes the polytope $\mathcal{P}_{\{\lambda_i\}}$, the proof of which is moved to Section A.2.

**Lemma 6.1** *The convex hull $\mathcal{P}_{\{\lambda_i\}}$ is a polytope with $2^k$ faces and $2^k$ corner points corresponding to the $2^k$ extremal columns of $S$. Further, the intersection of the manifold $\mathcal{M}_{\{\lambda_i\}}$ and the boundary of $\mathcal{P}_{\{\lambda_i\}}$ is only the set of those corner points. Hence, any point in the manifold is represented as a convex combination of the corner points, and it requires all the corner points to represent any point in the manifold that is not already one of the corner points.*

This implies that any column of $S$ can be represented as a convex combination of the extremal columns of $S$. We can write the dual of the primal LP in Equation (17) as:

$$
\underset{\mu \in \mathbb{R}^{2^k}}{\text{minimize}} \sum_{x \in \{0,1\}^k} \mu_x \quad \text{subject to}
$$

$$
\langle S_{(s_1,\ldots,s_k)}, \mu \rangle \ge \langle S_{(s_1,\ldots,s_k)}, W^{(y)} \rangle, \quad \text{for all } y \in \mathcal{Y}, (s_1,\ldots,s_k) \in [\lambda_1^{-1}, \lambda_1] \times \cdots \times [\lambda_k^{-1}, \lambda_k].
\tag{18}
$$

Consider an optimal dual solution $\mu^*$. We now prove that for any dual optimal solution, the constraints in Equation (18) can be met with equality only for the indices $(s_1,\ldots,s_k)$ corresponding to corner points of $\mathcal{P}_{\{\lambda_i\}}$. By complementary slackness of LP, this implies that the primal variable $\theta^{(y)}_{(s_1,\ldots,s_k)}$ can only be strictly positive for the extremal transcripts, and all non-extremal transcripts must be zero.

If $\langle T, \mu^* \rangle = \sum_x W_x^{(y)} T_x$ for some $T \in \mathcal{M}_{\{\lambda_i\}}$ which is not an extremal point, then it follows from Lemma 6.1 that $T$ can be represented as a convex combination of the extremal points. Unless all the constraints for $\mu^*$ are satisfied with equalities (which can only happen if $W^{(y)}$ are all same for all $y \in \mathcal{Y}$ and all protocols and decision rules achieve the same accuracy), there exists at least one extremal signature $S_{(s_1,\ldots,s_k)}$ such that the inequality in (18) is violated. Hence, it contradicts the assumption that $\mu^*$ is a feasible dual solution.

## 6.2 Proof for the worst-case accuracy

**Theorem 6.2** *For a central observer who wants to compute $f$ with accuracy measure $w$, randomized response with the optimal decision rule of (6) maximizes the worst-case accuracy $\mathrm{ACC}_{\mathrm{wc}}(P, w, f, \hat{f})$ among all $\{\lambda_i\}$-differentially private protocols and all decision rules.*

In this section, we provide a proof of this theorem. Consider the worst case accuracy of the form

$$
\min_{x \in \{0,1\}^k} \mathbb{E}_{\hat{f}(\tau)}[w(f(x), \hat{f}(\tau))] = \min_x \sum_{y \in \mathcal{Y}} \underbrace{w(f(x), y)}_{W_x^{(y)}} \sum_{\tau \in \mathcal{T}} P_{x,\tau} \underbrace{\mathbb{P}(\hat{f}(\tau) = y)}_{Q_{\tau,y}}.
$$

Using the signature matrix $S$, we can write this as maximizing a concave function (minimum over a set of linear functions is a concave function):

$$
\begin{aligned}
\underset{\theta^{(1)},\ldots,\theta^{(|\mathcal{Y}|)}}{\text{maximize}} \quad & \min_{x \in \{0,1\}^k} \left\{ \sum_{y \in \mathcal{Y}} W_x^{(y)} \left(S \theta^{(y)}\right)_x \right\} \\
\text{subject to} \quad & S \sum_{y \in \mathcal{Y}} \theta^{(y)} = \mathbb{1} \\
& \theta^{(y)} \ge 0.
\end{aligned}
\tag{19}
$$

This can be formulated as the following primal LP:

$$\underset{\xi, \theta^{(1)}, \ldots, \theta^{(|\mathcal{Y}|)}}{\text{maximize}} \quad \xi$$

$$\text{subject to} \qquad \xi \leq \left\{ \sum_{y \in \mathcal{Y}} W_x^{(y)} \left( S\theta^{(y)} \right)_x \right\}, \quad \text{for all } x \in \{0,1\}^k$$

$$S \sum_{y \in \mathcal{Y}} \theta^{(y)} = \mathbb{1} \tag{20}$$

$$\theta^{(y)} \geq 0.$$

Define dual variables $\nu \in \mathbb{R}^{2^k}$ corresponding to the first set of constraints and $\mu \in \mathbb{R}^{2^k}$ to the second. Then the dual LP is

$$\underset{\nu, \mu}{\text{minimize}} \quad \sum_x \mu_x$$

$$\text{subject to} \quad \langle S_{(s_1, \ldots, s_k)}, \mu \rangle \geq \sum_x W_x^{(y)} S_{x, (s_1, \ldots, s_k)} \nu_x, \quad \text{for all } y \in \mathcal{Y}, (s_1, \ldots, s_k) \in [\lambda_1^{-1}, \lambda_1] \times \cdots \times [\lambda_k^{-1}, \lambda_k]$$

$$\mathbb{1}^T \nu = 1$$

$$\nu \geq 0. \tag{21}$$

Consider an optimal solution $(\nu^*, \mu^*)$. This defines a polytope for the each column of $S$ put in a tensor form in $\mathbb{R}^{2^k}$:

$$\mathcal{P}_{\nu^*, \mu^*} = \{ T : \langle T, \mu^* \rangle \geq \sum_x W_x^{(y)} T_x \nu_x^*, \text{ for all } y \in \mathcal{Y} \}.$$

Now, $(\nu^*, \mu^*)$ is feasible if and only if $\mathcal{P}_{\{\lambda_i\}} \subseteq \mathcal{P}_{\nu^*, \mu^*}$, since the condition must be met by all $\lambda$-DP protocol-compatible transcripts.

Since both $\mathcal{P}_{\nu^*, \mu^*}$ and $\mathcal{P}_{\{\lambda_i\}}$ are convex polytopes, and $\mathcal{P}_{\{\lambda_i\}} \subseteq \mathcal{P}_{\nu^*, \mu^*}$ for our choice of optimal solutions, then the constraints in Eq. (21) can only be met with equality for signatures corresponding to the intersection of $\mathcal{M}_{\{\lambda_i\}}$ and the boundary of $\mathcal{P}_{\nu^*, \mu^*}$. From Lemma 6.1, we know that such intersection can only happen at the extremal points. By complementary slackness of LP, this implies that the primal variable $\theta_{(s_1, \ldots, s_k)}^{(y)}$ can only be strictly positive for the extremal transcripts, and all non-extremal transcripts must have zero value. However, in this case, one might need to resort to randomized decisions depending on the accuracy weights $W$.

The optimality of the extremal protocols can also be also explained perhaps more intuitively as follows. Consider the primal LP formulation. Let $\Theta = \{\theta^{(1)}, \ldots, \theta^{(|\mathcal{Y}|)}\}$ be an optimal solution that has at least one value that is non-extremal. Without loss of generality, let $\theta_i^{(1)}$ be the positive value corresponding to a non-extremal transcript $S_i$. Then, by the lemma, we know that we can represent $S_i = \sum_{j=1}^{2^k} \alpha_j S_j$, where $S_1 \ldots, S_{2^k}$ are the extremal transcripts. Then, we can construct another feasible solution $\tilde{\Theta} = \{\tilde{\theta}^{(1)}, \ldots, \tilde{\theta}^{(|\mathcal{Y}|)}\}$ from $\Theta$, by taking the value of $\theta_i^{(1)}$ and add it to the extremal ones according to $\tilde{\theta}_j^{(1)} = \theta_j^{(1)} + \alpha_j \theta_i^{(1)}$, and setting $\tilde{\theta}_i^{(1)} = 0$. The new solution preserves the summation $S\tilde{\theta}^{(y)} = S\theta^{(y)}$. Since the new solution has one less non-extremal value, we can continue in this fashion until we are left with only extremal transcripts.

# 7 Two-party function computation

In this section, we show that randomized response always dominates over any other non-interactive schemes. Precisely, we will show the following claim: *for any non-interactive protocol and a decision rule, there exists a randomized response and a decision rule for the randomized response that achieves the same accuracy, for any privacy level, any function, and any measure of accuracy.*

The statement is generally true, but for concreteness we focus on a specific example in the two-party setting, which captures all the main ideas. In this setting, there are essentially only two functions of interest, AND and XOR, and it is only interesting to consider the scenario where the central observer is trying to compute these functions over two bits distributed across two parties. Private AND function computation under the worst-case accuracy measure was studied in [22]. [22] proposed a non-interactive scheme and showed that it achieves the optimal accuracy of $\lambda(\lambda^2 + \lambda + 2)/(1 + \lambda)^3$ when both parties satisfy $\lambda$-differential privacy.

We will show by example how to construct a randomized response that dominates any non-interactive scheme. The protocol proposed in [22] outputs a privatized version of each bit according to the following rule

$$
M(0) = \begin{cases} 0 & \text{w.p.} \frac{\lambda}{1+\lambda} \\ 1 & \text{w.p.} \frac{\lambda}{(1+\lambda)^2} \\ 2 & \text{w.p.} \frac{1}{(1+\lambda)^2} \end{cases} \quad, \text{ and } \quad M(1) = \begin{cases} 0 & \text{w.p.} \frac{1}{1+\lambda} \\ 1 & \text{w.p.} \frac{\lambda^2}{(1+\lambda)^2} \\ 2 & \text{w.p.} \frac{\lambda}{(1+\lambda)^2} \end{cases},
$$

which satisfies $\lambda$-differential privacy. Such a non-interactive protocol of revealing the privatized data is referred to as a privacy mechanism. Upon receiving this data, the central observer makes a decision according to

$$
\hat{f}(M(x_1), M(x_2)) = \begin{cases} 1 & \text{if } M(x_1) = 2 \text{ or } M(x_2) = 2 \\ M(x_1) \wedge M(x_2) & \text{otherwise} \end{cases}.
$$

Now consider the randomized response mechanisms:

$$
M_{\mathrm{RR}}(x_i) = \begin{cases} x_i & \text{with probability} \frac{\lambda}{1+\lambda}, \\ \bar{x}_i & \text{with probability} \frac{1}{1+\lambda}. \end{cases}
$$

The dominance of this randomized response follows from a more general result proved in [34] which introduces a new operational interpretation of differential privacy mechanisms that provides strong analytical tools to compare privacy mechanisms.

This crucially relies on the following representation of the privacy guarantees of a mechanism. Given a mechanism, consider a binary hypothesis test on whether the original bit was a zero or a one based on the output of the mechanism. Then, the two types of errors (false alarm and missed detection) on this binary hypothesis testing problem defines a two-dimensional region where one axis is $P_{\mathrm{FA}}$ and the other is $P_{\mathrm{MD}}$. For a rejection set $S$ for rejecting the null hypothesis, $P_{\mathrm{FA}} = \mathbb{P}(M(x) \in S)$ and $P_{\mathrm{MD}} = \mathbb{P}(M(x) \notin S)$. The convex hull of the set of all pairs $(P_{\mathrm{MD}}, P_{\mathrm{FA}})$ for all rejection sets, define the hypothesis testing region. For example, the mechanism $M$ corresponds to region $\mathcal{R}_M$ and the randomized response corresponds to region $\mathcal{R}_{M_{\mathrm{RR}}}$ in Figure 1, which happens to be identical.

Figure 1: Three regions $\mathcal{R}_M$, $\mathcal{R}_{M_{\mathrm{RR}}}$, and $\mathcal{R}_\lambda$ are identical ($\varepsilon = 1.5$).

Differential privacy conditions can be interpreted as imposing a condition on this region:

$$
P_{\mathrm{FA}} + \lambda P_{\mathrm{MD}} \geq 1 , \quad \text{and} \quad \lambda P_{\mathrm{FA}} + P_{\mathrm{MD}} \geq 1 ,
$$

which defines a triangular region denoted by $\mathcal{R}_\lambda$ and shown in Figure 1.

**Theorem 7.1 ([34, Theorem 2.3])** *A mechanism is $\lambda$-differentially private if and only if the corresponding hypothesis testing region is included inside $\mathcal{R}_\lambda$.*

This is a special case of the original theorem which proves a more general theorem for $(\varepsilon, \delta)$-differential privacy. We can immediately check that both $M$ and $M_{\mathrm{RR}}$ are $\lambda$-differentially private.

It is no coincidence that the regions $\mathcal{R}_M$, $\mathcal{R}_{M_{\mathrm{RR}}}$, and $\mathcal{R}_\lambda$ are identical. It follows from the next theorem on the operational interpretation of differential privacy. We say a mechanism $M_1$ *dominates* a mechanism $M_2$ if $M_2(x)$ is conditionally independent of $x$ conditioned on $M_1(x)$. In other words, we can construct the following Markov chain: $x - M_1(x) - M_2(x)$. This is again equivalent to saying that there is another mechanism $T$ such that $M_2(x) = T(M_1(x))$. Such an operational interpretation of differential privacy brings both the natural data processing inequality (DPI) and the strong converse to the data processing inequality, which follows from a celebrated result of Blackwell on comparing two stochastic experiments [4]. These inequalities, while simple by themselves, lead to surprisingly strong technical results, and there is a long line of such a tradition in the information theory literature: Chapter 17 of [12] enumerates a detailed list.

**Theorem 7.2 (DPI for differential privacy [34, Theorem 2.4])** *If a mechanism $M_1$ dominates another mechanism $M_2$, then*

$$\mathcal{R}_{M_2} \quad \subseteq \quad \mathcal{R}_{M_1} \ .$$

**Theorem 7.3 (A strong converse to the DPI [34, Theorem 2.5])** *For two mechanisms $M_1$ and $M_2$, there exists a coupling of the two mechanisms such that $M_1$ dominates $M_2$, if*

$$\mathcal{R}_{M_2} \quad \subseteq \quad \mathcal{R}_{M_1} \ .$$

Among other things, this implies that among all $\lambda$-differentially private mechanisms, the randomized response dominates all of them. It follows that, for an arbitrary mechanism $M$, there is another mechanism $T$ such that $M(x) = T(M_{\mathrm{RR}}(x))$.

In the two-party setting, this implies the desired claim that there is no point in doing anything other than the randomized response, and that for the AND example, even though the protocol in [22] uses an alphabet of three letters for each party, it is still able to achieve maximum accuracy, because there is no reduction in the hypothesis testing region. The final decision is made as per $\hat{f}(M(x_1), M(x_2))$. Without doing any calculations, one could have guessed that this is achievable with randomized response which uses only the minimal two letters by simply simulating $M(x_i)$ upon receiving $M_{\mathrm{RR}}(x_i)$, namely, by computing $\hat{f}_{\mathrm{RR}}(M_{\mathrm{RR}}(x_1), M_{\mathrm{RR}}(x_2)) = \hat{f}(T(M(x_1)), T(M(x_2)))$. The new decision rule for randomized response is:

$$\hat{f}_{\mathrm{RR}}(M_{\mathrm{RR}}(x_1), M_{\mathrm{RR}}(x_2)) \quad = \quad \begin{cases} 0 & \text{if } (M_{\mathrm{RR}}(x_1), M_{\mathrm{RR}}(x_2)) = (0,0) \\ 1 & \text{if } (M_{\mathrm{RR}}(x_1), M_{\mathrm{RR}}(x_2)) = (1,1) \\ 0 & \text{if } (M_{\mathrm{RR}}(x_1), M_{\mathrm{RR}}(x_2)) = (0,1) \text{ or } (1,0), \text{ then with probability } \frac{\lambda}{1+\lambda} \\ 1 & \text{if } (M_{\mathrm{RR}}(x_1), M_{\mathrm{RR}}(x_2)) = (0,1) \text{ or } (1,0), \text{ then with probability } \frac{1}{1+\lambda} \end{cases} .$$

# 8 Generalization to Multiple Bits

As an example, consider the first party with one bit $x$ and the second party has two bits $y_1$ and $y_2$. Each bit needs to be protected as per $\varepsilon$-differential privacy. A central observer wishes to compute the following function:

$$f(x, y_1, y_2) \quad = \quad \begin{cases} y_1 \oplus y_2 & \text{if } x = 0 \ , \\ y_1 \wedge y_2 & \text{if } x = 1 \ . \end{cases}$$

Randomized response would publish privatized versions of $x$, $y_1$, and $y_2$ according to (9). In an interactive scheme, looking at $\tilde{x}$, the second party publishes (the privatized version of) either $y_1 \oplus y_2$ (if $\tilde{x} = 0$) or $y_1 \wedge y_2$ (if $\tilde{x} = 1$). Upon receiving the privatized data, the central observer makes optimal decisions in each case. Figure 2 illustrates how these two protocols compare in terms of average accuracy, where the accuracy is one if the approximation is correct and zero if the approximation is incorrect. For $\varepsilon = 0$, both protocols cannot do better than the best random guess of zero, which achieves average accuracy of $5/8 = 0.625$. For large $\varepsilon$, both protocols achieve the best accuracy of one.

Average accuracy                                   Average accuracy

Figure 2: Interactive protocols can improve over the randomized response, when each party owns multiple bits, for computing XOR or AND (left) and computing the Hamming distance (right).

Another example of multiple bit multi-party computation is studied in [31]. There are two parties each owning two bits of data $x \in \{0,1\}^2$ and $y \in \{0,1\}^2$, and a third party wants to compute the Hamming distance

$d_H(x, y) = \sum_{i=1}^{2} |x_i - y_i|$. Assuming each bit needs to be protected, the randomized response would reveal each bit via Equation 9. On the other hand, we can design an interactive scheme where one party reveals its two bits via the randomized response, and the other party then outputs its best estimate of the Hamming distance obeying differential privacy guarantees. Figure 2 illustrates how these two protocols compare in terms of average accuracy, where the accuracy is $2 - |d_H(x, y) - \hat{d}|$ where $\hat{d}$ is the optimal decision made by the third party; the Hamming distance $d_H$ is one if the approximation is correct and zero if the approximation is incorrect. For $\varepsilon = 0$, both protocols cannot do better than the best random guess of zero. which achieves average accuracy of $5/8 = 0.625$. For large $\varepsilon$, both protocols achieve the best accuracy of one.

# A   Appendix

## A.1   Proof of Corollary 3.1

Let $\tilde{X}$ denote the random output of the randomized response, and let $f(\tilde{X})$ denote the XOR of all $k$ bits. Notice that $P(X, \tilde{X}) = (\lambda^{k - d_h(X, \tilde{X})})/(1 + \lambda)^k$ where $d_h(\cdot, \cdot)$ denotes the Hamming distance. For a given $\tilde{X}$ the decision is either $f(\tilde{X})$ or the complement of it. We will first show that $f(\tilde{X})$ is the optimal decision rule.

It is sufficient to show that $\mathbb{E}[w(f(X), f(\tilde{X}))|\tilde{X}] \geq \mathbb{E}[w(f(X), \bar{f}(\tilde{X}))|\tilde{X}]$. Since, $\mathbb{E}[w(f(X), f(\tilde{X}))|\tilde{X}] = \sum_{i \text{ even}} \binom{k}{i} \lambda^{k-i}/(1 + \lambda)^k$ and $\mathbb{E}[w(f(X), \bar{f}(\tilde{X}))|\tilde{X}] = \sum_{i \text{ odd}} \binom{k}{i} \lambda^{k-i}/(1 + \lambda)^k$, it follows that

$$\mathbb{E}[w(f(X), f(\tilde{X}))|\tilde{X}] - \mathbb{E}[w(f(X), \bar{f}(\tilde{X}))|\tilde{X}] = (\lambda - 1)^k/(1 + \lambda)^k \geq 0 \,,$$

since $\lambda \geq 1$. By symmetry, the decision rule is the same for all $\tilde{X}$, and also for the worst case accuracy. This finishes the desired characterization of the optimal accuracy.

To get the asymptotic analysis of the accuracy, notice that $\mathbb{E}[w(f(X), f(\tilde{X}))] + \mathbb{E}[w(f(X), \bar{f}(\tilde{X}))] = 1$ and $\mathbb{E}[w(f(X), f(\tilde{X}))] + \mathbb{E}[w(f(X), \bar{f}(\tilde{X}))] = (\lambda - 1)^k/(1 + \lambda)^k = (e^\varepsilon - 1)^k/(2 + (e^\varepsilon - 1))^k = (1/2)^k \varepsilon^k + O(\varepsilon^{k+1})$. It follows that $\mathbb{E}[w(f(X), f(\tilde{X}))] = 1/2 + (1/2)^{k+1} \varepsilon^k + O(\varepsilon^{k+1})$.

## A.2   Proof of Lemma 6.1

Consider the following half space for $\mathbb{R}^{2^k}$. For an $a \in \{-1, +1\}^k$, the half space $H_a$ is defined as the set of $T \in \mathbb{R}^{2^k}$ satisfying

$$(-1)^k \left( \prod_{j \in [k]} a_j \right) \sum_{x \in \{0,1\}^k} \left( T_x \prod_{i \in [k]} (-\lambda_i)^{a_i \, x_i} \right) \geq 0 \,. \tag{22}$$

We claim that

$$\mathcal{P}_{\{\lambda_i\}} = \left\{ T \in \bigcap_{a \in \{-1, +1\}^k} H_a \,\Big|\, T_{0\ldots 0} = 1 \right\} \,.$$

It is straightforward to see that $\mathcal{M}_{\{\lambda_i\}}$ is inside the intersection of all $2^k$ half-spaces: all tensors in $\mathcal{M}_{\{\lambda_i\}}$ satisfy

$$(-1)^k \left( \prod_{j \in [k]} a_j \right) \prod_{i \in [k]} \left( 1 - \lambda^{a_i} t_i \right) \geq 0 \,,$$

for all $a \in \{-1, +1\}^k$. This immediately implies that the tensors satisfy (22). To show that it is indeed the convex hull, we need to show that $\mathcal{M}_{\{\lambda_i\}}$ intersects with the boundary of $\mathcal{P}_{\{\lambda_i\}}$ at every corner point. $\mathcal{P}_{\{\lambda_i\}}$ as defined above is $2^k - 1$ dimensional polytope in $2^k$ dimensional space, with at most $2^k$ faces and $2^k$ corner points. Each corner point is an intersection of $2^k - 1$ half spaces and the one hyperplane defined by $T_{0\ldots 0} = 1$.

Consider a corner point of $\mathcal{M}_{\{\lambda_i\}}$ represented by $a \in \{-1, +1\}^k$ as

$$T^{(a)} = [1, \lambda_1^{a_1}] \otimes \cdots \otimes [1, \lambda_k^{a_k}] \,.$$

It follows that $T^{(a)}$ is an intersection of $2^k - 1$ half spaces $H_b$ for $b \neq a$. Hence, every corner point of $\mathcal{P}_{\{\lambda_i\}}$ intersects with $\mathcal{M}_{\{\lambda_i\}}$. This finishes the proof.