[Reviews · NeurIPS 2015]

Submitted by Assigned_Reviewer_1

This paper looks at the multiparty computation problem and shows that if the computation must be done under differential privacy, each party should essentially use randomized response independently on its own data. The main approach is to use a linear program to model the problem. They must do a fair bit of work to characterize the optimum policy, which is randomized response.

This is a "heavy" review, although I do not have too many comments for this paper. I did not have adequate time to fully verify the proofs in the supplementary material.

1)

In (1) it might be worth clarifying again that the input has no probability distribution so this is not average over data sets. This is important because the discussion before (8) on protocol compatibility makes it seem like the input bits are in fact random.

2)

One issue I have with the manuscript is that the description of the analysis in the main document seems mostly like an advertisement for how hard and challenging it is without providing enough intuition or summary to really understand what is going on. Cramming a full journal paper into the NIPS format is hard, but I would encourage the authors to think of the reader who might want a little more formal vs. descriptive exposition. That is, there is no insight into the "novel transformation" mentioned before Section 3. It would be nice to see some of that transformation described in the main document.

3) Similarly how might one give more depth to the statement "[w]e need to utilize the convex geometry of the problem in order to show that interaction is not necessary"?

In the end is this geometric insight intuitive or counterintuitive?

4) The fact that the optimality of non-interaction holds only for single bits is, to my mind, a major drawback of this work, and is in fact rather hidden in the exposition in the beginning of the paper, where a highlighted statement "non-interactive randomized response is exactly optimal" is made that seems to be very general. This is a kind of false advertising, since the single-bit case is made to seem like a toy example that contains the insights as opposed to perhaps a pathological example where parties have extremely limited information and few options. To be clear, I think this is an interesting result, but I think one should present it plainly, warts and all.

5) Finally, the main concern I have with this paper is the fit with NIPS. This is fundamentally a paper about MPC and not really about learning, machine or otherwise. While there are some intersections, this feels very much like a TCS paper and may be of more interest, more impactful, and better received in a venue such as NIPS. That being said, I think it's a solid and interesting piece of work.

Smaller issues: p2: The Sweeney paper [37] did not reveal the medical record of the governor. Please read the paper and state more accurately what was actually disclosed. p5-6: The sentence going across the page is not grammatical. p6: "but it turns out" p6: "the the" --> "the"

AFTER AUTHOR FEEDBACK: I thank the authors for their responses. I think my score stands as-is. I'm still not convinced about the fit, especially given the limitation to 1-bit protocols (and indications that it does not extend to multi-bit protocols). I don't think the audience at NIPS will be as appreciative.
Summary: This paper says that to do multiparty computation under differential privacy, it is optimal for each party to randomize its own data independently when each party only has a single bit.

Submitted by Assigned_Reviewer_2

The paper addresses a problem of multi-party computation under differential privacy. The main result is the exact optimality of a simple non-interactive protocol in terms of maximizing accuracy for any given privacy level, which requires that each party randomizes and publishes its own bit.

The paper is in general well-written and the main result is interesting. A minor comment is that it is much better to combine Subsection 3.1 into Sec. 3, that is, do not have Subsec. 3.1 as a single subsection.

Summary: The paper shows the optimality of a simple non-interactive protocol, which says that the simple randomized response is the optimal protocol in a very general sense.

Submitted by Assigned_Reviewer_3

Summary of paper: The paper analytically studies the optimality of randomized response for differentially private multiparty computation of general functions using general accuracy measures.

Quality: The paper present a concrete problem in privacy and security, and present stong results with careful proofs.

Clarity: The paper is written clearly overall, and the main results are well presented.

However, the proof of the key theorem is all shoved under the supplementary material, and it would have been nicer to see a sketch of the proof and intuitions behind the theory in the main paper.

Originality: While some components of the proposed setting and solutions are built on related work cited in the paper, the main results seem new and interesting.

Significance: Differentially private SMC has received attentions recently, and the results of the paper is very timely. Despite some restrictions mentioned in Discussion, the paper will appeal to the interested audience. However, since the topic of the paper is almost purely cryptography/security/privacy, I'm not sure if NIPS is the best venue for publishing this paper.

typo : line 270.
Summary: The paper present concrete results for a small but fundamental problem in cryptography/security/privacy back up by careful proofs.

Author Feedback
Author rebuttal: We would like to thank the reviewers for their detailed comments.

Reviewer_1:
1) We thank the reviewer for pointing this out, and we will clarify this in the final version.
2) We will give more formal exposition of the proof ideas and the challenges involved.
3) The statement could be better stated, by saying that "we define a linear program relaxation with an infinite set of linear constraints that include the feasible set. Fortunately, the optimal solution of this LP relaxation is unique and is also feasible in the original problem. Hence, we found the optimal solution of the original problem, and this is the non-interactive randomized response."
4) We will present in plain language that our main contribution is in proving the optimality of 1-bit multi party computation.
5) We believe that multiparty computation is an important topic in distributed learning/inference/decision making. We bring a new angle of data privacy into this distributed learning setting, which has potential of being of a broader interest to the NIPS community.
6) We will revise the smaller issues.

Reviewer_2:
Regarding combining Section 3.1: We agree that those sections are better combined, and will do so in the final version.

Reviewer_3:
Regarding clarity: We will add a high-level proof idea in the main text.
Regarding the venue: We believe that multiparty computation is an important topic in distributed learning/inference/decision making. We bring a new angle of data privacy into this distributed learning setting, which has potential of being of a broader interest to the NIPS community.

Reviewer_4:
Regarding novelty of the contribution :
Optimality of the randomized response has been shown in local differential privacy context, where there is no interaction between those parties. However, it is not at all clear that interaction does not help in multi party computation setting. We show for the first time that when restricted to Boolean data, there is no gain in interaction. Fundamentally, allowing for interaction amongst users complicates the analysis considerably and requires new mathematical techniques.